# Epigenetic Modifiers in Cancer Metastasis

**DOI:** 10.3390/biom14080916

**Published:** 2024-07-27

**Authors:** Die Hu, Tianci Zhao, Chenxing Xu, Xinyi Pan, Zhengyu Zhou, Shengjie Wang

**Affiliations:** 1Key Laboratory of Molecular Genetics between Kangda College of Nanjing Medical University and Suzhou Medical College of Soochow University, Suzhou 215123, China; hudie4560@163.com; 2Department of Basic Medicine, Kangda College of Nanjing Medical University, Lianyungang 222000, China; isacxu@126.com (C.X.); panxy031415@foxmail.com (X.P.); 3Key Laboratory of Cell Biology, Ministry of Public Health and Key Laboratory of Medical Cell Biology, Ministry of Education, China Medical University, Shenyang 110122, China; 2023110454@cmu.edu.cn; 4Laboratory Animal Center, Suzhou Medical College of Soochow University, Suzhou 215123, China

**Keywords:** epigenetics, DNA methylation, histone methylation, non-coding RNAs, metastasis

## Abstract

Metastasis is the primary cause of cancer-related death, with the dissemination and colonization of primary tumor cells at the metastatic site facilitated by various molecules and complex pathways. Understanding the biological mechanisms underlying the metastatic process is critical for the development of effective interventions. Several epigenetic modifications have been identified that play critical roles in regulating cancer metastasis. This review aims to provide a comprehensive summary of recent advances in understanding the role of epigenetic modifiers, including histone modifications, DNA methylation, non-coding RNAs, enhancer reprogramming, chromatin accessibility, and N6-methyladenosine, in metastasis-associated processes, such as epithelial-mesenchymal transition (EMT), cancer cell migration, and invasion. In particular, this review provides a detailed and in-depth description of the role of crosstalk between epigenetic regulators in tumor metastasis. Additionally, we explored the potential and limitations of epigenetics-related target molecules in the diagnosis, treatment, and prognosis of cancer metastasis.

## 1. Introduction

Cancer is characterized by uncontrolled cell proliferation that leads to the activation of oncogenic pathways that evade endogenous control [1]. Metastasis, the spread of cancer cells to other parts of the body, is a key stage in cancer progression and is responsible for at least two-thirds of all cancer-related deaths [2]. This process occurs through the bloodstream or lymphatic system and invades other organs and tissues [3,4]. Cancer metastasis is a multi-step biological process called the invasion–metastasis cascade, which is a hallmark of cancer progression [5]. Metastasis is a “hidden” process that occurs within the body and is difficult to detect [6]. Therefore, there is a need to elucidate the molecular mechanisms underlying tumor metastasis.

Despite the clinical importance of metastasis, much remains unknown regarding the molecular mechanisms underlying the metastatic process. Recent studies have identified the involvement of various epigenetic modifiers such as DNA methylation, histone modifications, non-coding RNAs, enhancer reprogramming, chromatin accessibility, and N6-methyladenosine in the proliferation and metastasis of cancer cells. These epigenetic modifiers have been shown to affect various signaling pathways, regulate transfer-related factors and proteins, and modulate the activity of transcription factors, suggesting that they are important therapeutic targets for cancer metastasis [7,8,9,10].

Therefore, in this review, we provide an overview of the molecular mechanisms and activities of epigenetic modifiers associated with cancer metastasis, focusing on their potential as therapeutic targets and diagnostic indicators for cancer (Figure 1). In addition, the mechanism of complex crosstalk between epigenetic regulators in tumor metastasis is illustrated. We also discuss the recent clinical advancements in epigenetic therapy, highlighting the potential of these therapies to improve cancer treatment outcomes.

## 2. Characteristics of Cancer Metastasis

Metastasis is the leading cause of poor prognosis and death among patients with cancer. A diagnosis of metastasis typically involves imaging studies, laboratory tests, and tissue biopsies. The treatment of metastatic cancer is typically challenging because of the need for a comprehensive approach that considers the extent and location of the metastases. Non-metastatic tumors detected at an early stage are usually cured with surgery, radiotherapy, or chemotherapy [11]. However, metastatic cancer is more difficult to treat. Metastatic cancer cells often exhibit resistance to therapies that are effective against primary tumors. Additionally, many patients who do not show evidence of metastasis at the time of the initial diagnosis may develop metastases at a later stage. Alternatively, metastases may occur years or decades after the apparent success of initial treatment [12,13]. Therefore, understanding the characteristics of cancer metastasis is crucial for early detection, accurate diagnosis, and effective disease management.

Cancer metastasis is a complex process with several distinguishing characteristics. First, cancer cells can invade the surrounding tissues by penetrating the boundaries of the original tumor and gaining access to adjacent structures. Second, these cells can enter the bloodstream or lymphatic vessels, enabling them to travel throughout the body and reach distant sites. During metastasis, cancer cells can survive in the circulatory system and adapt to new environments. They may develop mechanisms to evade the body’s immune system. Upon reaching a distant location, cancer cells can exit the blood vessels and establish secondary tumors. These secondary tumors are often referred to as metastatic tumors or metastases [14,15,16,17]. Metastasis often leads to various complications including damage to vital organs and tissues, resulting in functional impairment. Cancer metastasis involves a series of sequential interrelated steps. Each of these steps can be rate-limiting, as failure or inadequacy of any of the steps can prevent the entire course. This process is influenced by both the intrinsic properties of tumor cells and the host response. Cancer metastasis is not a random process, and certain tumors appear to have organ-specific metastasis patterns, with a tendency to preferentially metastasize to specific organs [18]. Cancer cells and other factors in the organ environment contribute to this organ specificity. Furthermore, not all cancer cells exhibit the same metastatic potential. A subset of cells, often referred to as cancer stem cells (CSCs), can initiate metastasis. These cells undergo clonal selection, in which the fittest and most invasive cells survive and proliferate in the new environment, ultimately resulting in the formation of metastatic lesions [19].

Epithelial-to-mesenchymal transition (EMT) is one of the initial and key mechanisms leading to cancer metastasis. During EMT, epithelial cells, which are the primary cells of a tumor, undergo changes in their molecular and cellular characteristics to adopt a mesenchymal phenotype. EMT results in cancer cells with stem-cell-like features that tend to invade surrounding tissues and exhibit resistance to certain therapeutic interventions. Given the clinical significance of EMT in cancer progression, the inhibition of EMT is a promising therapeutic approach that may have a significant impact on disease outcomes [20,21].

In summary, cancer metastasis is an evolving disease that is a combined result of cellular metastasis and a range of microenvironmental factors. Understanding these mechanisms is critical for developing new therapeutic strategies to prevent or treat metastatic diseases. Future research should focus on preventing and overcoming cancer metastasis.

## 3. Epigenetic Alterations in Cancer Metastasis

Epigenetic modifications are heritable changes in gene expression that do not affect the underlying DNA sequence [22]. Epigenetic regulation is achieved through various mechanisms including DNA methylation, histone modification, non-coding RNAs, chromatin accessibility, enhancer reprogramming, and N6-methyladenosine (m6A). These epigenetic alterations are critically associated with various mechanisms of proliferation and metastasis in several types of cancer.

### 3.1. DNA Methylation

Since its discovery in bacteria in 1925, DNA methylation has been linked to various biological functions, including gene regulation, genomic organization, reproduction and development, disease, and aging [23,24,25].

The following is a brief description of the key elements involved in the DNA methylation process—DNA methyltransferase, methylated CpG adhesion protein, and Tet enzymes (Ten-eleven translocation enzymes). DNA methyltransferases are a class of enzymes capable of adding methyl (-CH) groups to specific bases of DNA molecules [26]. According to their different functions, this is mainly divided into two categories. DNMT1, maintenance methyltransferase, mainly occurring in the process of DNA replication, with the newly synthesized daughter DNA corresponding to the parent chain methylation site of cytosine (C) methylation modification in order to maintain the stable inheritance of the DNA methylation pattern [27]. DNMT3A and DNMT3B, de novo methyltransferases that can catalyze new methylation at the unmethylated CpG site and establish the initial DNA methylation pattern [28]. DNA methyltransferase activity is regulated by a variety of factors, including interactions with other proteins, intracellular metabolites, transcription factors, and chromatin structure. Methylated CpG adhesion proteins are a class of proteins that can specifically recognize and bind to methylated CpG sites. By binding to methylated DNA, they recruit other protein complexes, such as histone-modifying enzymes, chromatin-remodeling factors, etc., to form inhibitory chromatin structures, thereby inhibiting gene transcriptional expression [29]. Tet enzymes are a class of dioxygenases that catalyze the oxidation of 5-methylcytosine (5mC) to 5-hydroxymethylcytosine (5hmC), 5-aldehyde cytosine (5fC), and 5-carboxycytosine (5caC). The activity of the Tet enzyme is essential for the process of DNA demethylation. The 5hmC, 5fC, and 5caC catalyzed by them can be removed by the Base Excision Repair (BER) pathway or by deamination, thus achieving DNA demethylation [30].

DNA methylation displays a distinct distribution pattern across the genome. CpG islands are genomic regions abundant in cytosine (C)-guanine (G) dinucleotides, primarily located in gene promoter regions, most of which remain unmethylated under normal physiological conditions, thereby facilitating the expression of associated genes [31]. The CpG shore refers to the region located within approximately 2 kb upstream and downstream of the CpG Island. In certain instances, alterations in the methylation status of the CpG shore also impact the transcriptional activity of neighboring genes [32]. TEs and other forms of repetitive DNA, constituting more than half of the human genome, are the largest contributors to human genetic variation and affect human health owing to their roles in deleterious copy number variants (CNVs), structural variants (SVs), insertions, deletions, and alterations to gene transcription and splicing [33,34]. DNA methylation plays a crucial role in repetitive genomic regions such as retrotransposons by methylating these regions and repressing their transcription and transposition activities, thus maintaining genomic stability. At Pancreatic ductal adenocarcinoma (PDAC), Espinet found that hypomethylated tumors are characterized by a higher expression of delivered endogenous transcripts and dsRNA sensors, which lead to cell-intrinsic activation of an interferon signature (IFNsign). This results in a pro-tumorigenic microenvironment and poor patient outcome [35].

Aberrant changes in DNA methylation distribution patterns occur in cancer, including hypermethylation in the tumor-suppressor gene promoter region, CpG island, and hypomethylation in repeat sequence regions. These modifications significantly affect gene expression and genome stability, exerting a critical influence on cancer initiation, progression, and metastasis [36,37] (Figure 2). In particular, abnormal DNA methylation is crucial for cancer metastasis. Itoh et al. reported that interleukin-6 (IL-6) expression in primary tumor cells is increased by the hypoxic tumor microenvironment and that increased expression may contribute to tumor metastasis. IL-6 expression in tumor cells is regulated by DNA methylation. In primary tumor cells, IL-6 expression is increased by DNA demethylation of its promoter via ten-eleven translocation 2 (Tet2). Tet2 activity is associated with increased metastatic capacity of OS cells. Tet2-dependent IL-6 induction promotes a metabolic shift and increased lung colonization activity, enhancing metastasis in osteosarcoma (OS) cells. These findings suggest that blocking IL-6 signaling could serve as a potential therapy to antagonize metastasis [38]. Gallon et al. provided new evidence on the DNA methylation characteristics and potential mechanisms of prostate cancer brain metastases (PCBMs). Abnormal methylation of PCBMs is related to mutational background and polycomb repressive complex 2 (PRC2) complex activity. While PCBMs showed a hypermethylated phenotype of CpG islands, the gene promoters involved in neuroactive ligand–receptor interactions and cell adhesion molecules such as *GABRB3*, *CLDN8*, and *CLDN4* were hypomethylated, suggesting that primary tumor cells may require specific reprogramming to develop brain metastases [39].

EMX1 was recognized as a potential clinically available epi-marker in hepatocellular carcinoma (HCC) [40]. Wen et al. found that EMX1 gene body DNA methylation at R2–R4 positively regulates EMX1 expression in HCC. EMX1-FL, but not EMX1-X1, promotes the migration, invasion, and metastasis of HCC in vivo and in vitro by activating *EGFR* transcription and *EGFR-ERK* signaling, while blocking the *EGFR* signals can reverse these effects and reduce HCC metastasis. Their results revealed that *EMX1-EGFR* is a potential target for reducing HCC metastasis [41]. Bao-Caamano also validated the epigenetic regulation of relevant genes in the human colorectal cancer (CRC)-derived cell line CTC-MCC-41. The methylation profile of CTC-MCC-41 was completely different from that of primary and metastatic CRC cells, which were distinguished by a slight predominance of hypomethylated CpGs located primarily in CpG-poor regions. The methylation profile of the promoter CpG islands and shore regions of CTC-MCC-41 is associated with the transcriptional program and relevant cancer pathways, primarily *Wnt* signaling [42]. Repetitive elements (REs) compose about 50% of the human genome and are normally transcriptionally silenced [43], although the mechanism has remained elusive. Shen et al. identified FBXO44 as an essential repressor of REs in cancer cells. FBXO44 bound H3K9me3-modified nucleosomes at the replication fork and recruited SUV39H1, CRL4, and Mi-2/NuRD to transcriptionally silence REs post-DNA replication. FBXO44/SUV39H1 are crucial repressors of RE transcription, and their inhibition selectively induces DNA replication stress and viral mimicry in cancer cells [44]. Collectively, aberrant DNA methylation status may represent a potential biomarker of cancer and a promising therapeutic target for cancer metastasis, as summarized in Table 1.

### 3.2. Histone Modification

Histone modifications, including methylation, acetylation, ubiquitination, and phosphorylation, are critical for regulating oncogenes and tumor-suppressor genes at the transcriptional level during tumor progression [58]. Recent studies have revealed a strong correlation between histone modification and the expression of genes associated with cancer metastasis (Table 2).

Chromodomain helicase DNA-binding protein 5 (CHD5) is recognized as a tumor-suppressor gene in cancer that controls proliferation, apoptosis, and senescence via the *p19(Arf)/p53* pathway [71]. Xie et al. reported that CHD5 is epigenetically silenced in HCC via *PRC2*-mediated histone H3 trimethylation at lysine 27 (H3K27me3). Research has indicated that CHD5 inhibits the migration and invasion of HCC cells. In addition, there is a reciprocal inhibitory regulation between EZH2 and CHD5 in HCC tissues, which may have implications for their potential therapeutic significance in HCC treatment [72]. Azevedo et al. found that clusters of histone modifications were associated with melanoma progression, EMT, and metastasis. Melanoma onset was linked to the reduced acetylation of H4K5 and H4K8, whereas EMT was associated with methylation levels of H3K4, H3K56, and H4K20 [73]. Similarly, Chen et al. showed that lysine-specific histone demethylase 5C (KDM5C)-mediated demethylation of histone H3 lysine 4 trimethylation (H3K4me3) in the promoter of methyltransferase-like 14 (METTL14) inhibited *METTL14* transcription. Moreover, they confirmed that METTL14 inhibited the migration, invasion, and metastasis of CRC cells using in vitro and in vivo assays, respectively [61].

Additionally, the role of histone acetylation in cancer metastasis is well documented. For instance, Hu et al. found that histone deacetylase inhibitors (HDACi) promoted breast cancer metastasis by elevating NEDD9 expression. *NEDD9*, also named *casl* and *hef1*, encodes a multi-domain scaffolding protein involved in cell signaling and is known to regulate multiple cellular processes, such as cell proliferation, DNA damage response, migration, and so on [74,75]. Some studies have confirmed that NEDD9 stimulates breast cancer cells invasion by influencing EMT and activating MMP [76]. Their data demonstrated that pan-HDAC inhibitors enhanced H3K9 acetylation at the *NEDD9* gene promoter by inhibiting the HDAC4-stimulated *NEDD9-FAK* signaling pathway, leading to breast cancer cell metastasis. Notably, *FAK* inhibitors reversed breast cancer metastasis induced by NEDD9 upregulation via pan-HDAC inhibitors. These results suggest a potential combination therapy for breast cancer [63]. Similarly, Zhang et al. reported that upregulation of collagen type VI alpha 1 (*COL6A1*) through H3K27 acetylation promoted OS lung metastasis by suppressing the *STAT1* pathway and activating pulmonary cancer-associated fibroblasts. *COL6A1* plays a role in cell migration, differentiation, embryonic development, and the maintenance of cell stemness [77]. Their study revealed a novel molecular mechanism for OS cell metastasis to the lungs, as well as crosstalk between tumor cells and fibroblasts, contributing to the development of effective prevention and treatment strategies for OS [62].

Histone modifications such as ubiquitination and acetylation play a crucial role in cancer metastasis. RNF2 (also termed *Ring 1B*), a major component of polycomb repressive complex 1 (PRC1), is a type of RING domain-containing E3 ubiquitin ligase [78]. Yao et al. indicated that RNF2 contributes to HCC metastasis via inducing histone mono-ubiquitination, suggesting that *RNF2* could be a potential therapeutic target for HCC. Mechanistically, RNF2-regulated crosstalk between H2AK119ub, H3K27me3, and H3K4me3 synergistically represses *E-cadherin* transcription and promotes EMT and HCC metastasis. Another important histone modification is histone phosphorylation, which has been studied in the progression of cancer metastasis [68]. Li et al. revealed that inhibiting the *HECTD3-IKKα* axis effectively suppressed tumor metastasis induced by systemic inflammation, providing a potential strategy for the prevention and treatment of tumor hematogenous metastasis. HECTD3 is a HECT-type E3 ubiquitin ligase with multiple substrates and functions [79]. HECTD3 promotes the stabilization, nuclear localization, and kinase activity of IKKα by ubiquitinating IKKα with K27- and K63-linked polyubiquitin chains at K296, increasing histone *H3* phosphorylation, and promoting *NF-κB* target gene transcription [69].

### 3.3. Non-Coding RNAs

Non-coding RNAs (ncRNAs), which constitute the vast majority of the human transcribed genome, serve a multitude of functions in crucial biological processes and are associated with various pathological conditions, particularly cancer. Regulatory ncRNAs can be categorized as microRNA (miRNA), small interfering RNA (siRNA), piwi-interacting RNA (piRNA), long non-coding RNA (lncRNA), and circular RNA (circRNA) [80]. In recent years, growing evidence has suggested that miRNAs, lncRNAs, and circRNAs play a significant role in the regulation of tumor metastasis (Table 3).

MicroRNAs (miRNAs) are ncRNAs with lengths ranging from approximately 19 to 22 nucleotides that regulate gene-expression post transcriptionally. They regulate gene expression mainly by targeting specific recognition sites in the 3′ untranslated region (UTR), leading to mRNA degradation [95]. Dysregulation of miRNAs is associated with the development and progression of cancer. For example, in gastric cancer, *miR-20a-5p* and *miR-221* are elevated in both plasma and serum samples. Changes in *miR-221-3p*, *miR-378c-3p*, and *miR-744-5p* in serum can aid the prediction of gastric cancer 5 years before any clinical symptoms appear [96]. Furthermore, the expression of *miRNA-21-5p* in plasma can identify colorectal cancer patients with 90% sensitivity and specificity [97]. Notably, they play pivotal roles in the regulation of multiple cellular functions, including cell proliferation and migration, and EMT phenotype. miRNAs have emerged as potential therapeutic targets for the progression of various tumor metastases.

*MicroRNA-122 (miR-122)* is a liver-specific miRNA that regulates various hepatic functions [98]. Sendi reported the development of a nanoformulation of *miR-122* as a therapeutic agent to prevent liver metastasis. Galactose-targeted lipid calcium phosphate (Gal-LCP) nanoformulations of *miR-122* can selectively deliver *miR-122* to hepatocytes with high efficiency, effectively preventing CRC liver metastasis and prolonging survival. Mechanistically, Gal-LCP *miR-122* therapy has been associated with the downregulation of key genes involved in metastatic and cancer inflammatory pathways as well as an increase in the CD8+/CD4+T cell ratio and a decrease in immunosuppressive cell infiltration [99]. Furthermore, Han suggested that silencing *miR-106a* to restore tumor protein 53-induced nuclear protein 1 (TP53INP1) expression could be a novel therapeutic strategy for bone metastasis in lung adenocarcinoma. *miR-106a* levels are elevated in lung cancer patients with bone metastasis. Mechanistic investigations have revealed that *miR-106a* upregulation promotes metastasis by targeting TP53INP1-mediated metastatic progression, including cell migration and EMT [100].

Long non-coding RNAs (lncRNAs) are a group of RNA molecules with lengths greater than 200 nucleotides. A majority of lncRNAs are transcribed by RNA polymerase II and matured by 5′ capping, 3′ cleavage, and polyadenylation and splicing [101]. Unlike mRNAs, few lncRNAs are universally expressed in tissues. Instead, most are expressed specifically in specific conditions and tissues, meaning they have functional relevance [102,103,104]. They play a key role in the regulation of cellular processes such as genome integrity, chromatin organization, gene expression, translation regulation, and signal transduction [105]. Compared to miRNAs, lncRNAs exhibit more diverse mechanisms for regulating gene expression [106]. In recent years, lncRNAs have been implicated in various biological processes, including tumor proliferation, migration, invasion, and angiogenesis [107]. In addition, there is evidence that exosomes are closely related to tumor metastasis, and exosomes can arrive at the site of metastasis before tumor cells, change the microenvironment, activate the biological function of cells, and thus promote tumor metastasis [81]. Exosomes contain a large number of molecules including lncRNAs. This protects lncRNAs from degradation [108]. In this review, we have outlined the molecular mechanisms by which lncRNAs contribute to cancer metastasis.

Metastasis-associated lung adenocarcinoma transcript1 (MALAT1) is a conserved lncRNA in mammals that is highly abundant in many cancers [109]. Transforming growth factor beta (TGF-β)-induced upregulation of MALAT1 enhances cancer metastasis, which is mediated by an interaction with suz12 at the transcriptional level to alter downstream events. Instead, MALAT1 silencing reduces cell proliferation and invasion and increased apoptosis by reducing MALAT1-EZH2-mediated target silencing at the epigenetic level [110]. Ni reported that exosomal *lncRNA-SOX2OT* contributes to bone metastasis (BoM) and may represent a potential therapeutic target for metastatic NSCLC. NSCLC cell-derived exosomal *lncRNA-SOX2OT* promoted cell invasion and migration in vitro and in vivo. These findings highlight a molecular mechanism whereby *lncRNA-SOX2OT* modulates osteoclast differentiation and stimulates BoM by targeting the *miRNA-194-5p/RAC1* signaling axis and the TGF-β/*pTHrP*/*RANKL* signaling pathway in osteoclasts [111]. Chen also demonstrated the significance of lncRNA in cancer metastasis. They found that SUMOylation promotes extracellular vesicle-mediated transmission of *lncRNA ELNAT1* and lymph node (LN) metastasis in bladder cancer (BCa). Mechanistically, the extracellular vesicle-mediated *ELNAT1/UBC9/SOX18* regulatory axis promotes lymphangiogenesis and LN metastasis in BCa via SUMOylation. These findings suggest that the lncRNA ELNAT1 may serve as a promising prognostic biomarker and therapeutic target for LN metastatic BCa [85].

Circular RNAs (circRNAs) are characterized by covalently closed loops generated through a process called back-splicing, resulting in circular structures that lack 5′ caps and 3′ poly (A) tails [112]. Recent studies on circRNAs in cancer have revealed that they are significantly involved in cancer initiation, proliferation, drug resistance, and most importantly, cancer metastasis [113,114,115,116]. One of the primary mechanisms by which circRNAs influence tumor metastasis is through their regulation of gene expression at the transcriptional or post-transcriptional level. For instance, circRNAs bind to miRNAs through a regulatory mechanism in which endogenous RNAs compete to indirectly regulate the expression of mRNA corresponding to downstream target genes of miRNAs, contributing to the progression of breast cancer [117]. In addition, circRNAs can also interact with proteins to affect their functions and signaling pathways related to tumor metastasis. For example, certain circRNAs can bind to proteins involved in EMT. Through these interactions, circRNAs can regulate EMT and subsequently impact the metastatic behavior of tumor cells [118]. Furthermore, circRNAs can be secreted into the extracellular space and enter the circulation, functioning as potential biomarkers for tumor metastasis. The detection of specific circRNAs in the blood or other body fluids of cancer patients may provide valuable information for the early diagnosis and prognosis of metastatic tumors, as well as for monitoring the response to therapy [119].

In recent years, circRNAs have emerged as crucial regulators in tumor progression and metastasis, drawing significant attention from the scientific community. Chen found that *circRHOBTB3* exerts suppressive effects on CRC aggressiveness through the human antigen R (HuR)/polypyrimidine tract-binding protein 1 (PTBP1) axis. Mechanistically, *circRHOBTB3* binds to *HuR* and promotes the *β-Trcp1*-mediated ubiquitination of *HuR*, leading to *HuR* degradation and diminished expression of the downstream target PTBP1. As reported by Takahashi et al., PTBP1 plays a crucial role in the invasion of CRC cells, and these invasive properties arise partially through splicing CD44 [120]. Moreover, PTBP1 has been reported to mediate the pyruvate kinase (PKM) isotype switch to facilitate the Warburg effect in CRC [121]. Chen also found that the overexpression of *circRHOBTB3* represses *PTBP1*-mediated CRC metastasis [122]. Du revealed that *circNFIB (cNFIB)* inhibits the proliferation and metastasis of intrahepatic cholangiocarcinoma cells (ICCs) in vitro and in vivo. The underlying mechanism involves the competitive interaction of *cNFIB* with mitogen-activated protein kinase kinase1 (MEK1), resulting in the dissociation of MEK1, and mitogen-activated protein kinase 2 (ERK2), ultimately suppressing *ERK* signaling and tumor metastasis. Furthermore, overexpression of exogenous *cNFIB* enhanced the anti-tumor effects of trametinib (a specific MEK inhibitor), suggesting its enormous potential as a therapeutic candidate for combating ICC metastasis [123].

### 3.4. Other Epigenetic Modifiers

Epigenetic regulators have been linked to several pathways that are involved in cancer metastasis. In addition to DNA methylation, histone modification, and ncRNAs, enhancer reprogramming, chromatin accessibility, and m6A have emerged as potential drivers of cancer metastasis in various malignancies (Table 4).

Enhancers are cis-regulatory elements that play key roles in controlling specific gene expression programs over long distances, ultimately determining cell type or cellular state [134]. Roe reported an association between FOXA1 and PDA metastasis, which was unexpected because previous studies have shown that FOXA1 is expressed at lower levels in mesenchymal PDA cell lines [135]. Further investigations revealed that FOXA1-mediated enhancer reprogramming enhanced metastatic potential. This observation is consistent with the measurement of FOXA1 expression in both mouse and human PDA samples [132]. Li demonstrated that enhancer reprogramming is accompanied by cancer-associated fibroblast (CAF) activation during cancer progression. Specifically, they found that activated *JUN* in the stroma was both essential and sufficient to remodel the CAF-specific enhancer landscape. It promotes the expression of pro-metastatic genes and augments breast cancer invasiveness. These cases collectively demonstrate that enhancer reprogramming is a key driver of changes in the cell type and function [136].

Physical access to DNA is a highly dynamic chromatin property that is essential for establishing and maintaining cellular identity. The organization of accessible chromatin across the genome reflects a network of permissible physical interactions [137]. A growing body of evidence implicates chromatin accessibility in the development and progression of cancer [138,139,140]. For example, Pierce identified LKB1 as a key regulator of chromatin accessibility in primary lung adenocarcinoma tumors. They found that LKB1 deletion stimulates the activation of the early endoderm transcription factor SOX17 in metastases and a metastasis-like sub-population of cancer cells within primary tumors. SOX17 expression is necessary and sufficient to trigger a second wave of epigenetic modifications in LKB1-deficient cells, which in turn enhances their metastatic ability [141].

M^6^A is a prevalent internal modification of mRNAs in eukaryotes and has been linked to diverse effects on mRNA fate [142]. It is involved in almost all RNA cycle stages, including the regulation of transcription, maturation, translation, degradation, and stability of mRNA [143]. Additionally, m6A has been implicated in influencing tumor migration and invasion. Li revealed that METTL3 enhances cell migration, invasion, and EMT by regulating the expression and membrane localization of β-catenin (encoded by *CTNNB1*) in various types of cancer cells, including cervical, lung, and liver cancers. Specifically, METTL3 regulates β-catenin transcription, mRNA degradation, translation, and subcellular localization. For CTNNB1 expression, METTL3 indirectly inhibits *CTNNB1* transcription by stabilizing its transcriptional suppressor, *E2F1* mRNA, and promoting *CTNNB1* inhibition via YTHDF2 recognition-dependent modification of the 5′-UTR m6A. M6A modification in the 5′-UTR region inhibits the translation efficiency of *CTNNB1*, although the overall level of METTL3 affects both the canonical and non-standard translation of *CTNNB1*, possibly via the interaction of YTHDF1 with eIF4E1/eIF4E3. Moreover, METTL3 downregulates c-Met kinase to inhibit β-catenin membrane localization and its interaction with E-cadherin, resulting in the inhibition of cell movement [144]. Yin et al. found that RNA m6A methylation regulates tumor growth and metastasis via macrophage reprogramming. Specifically, the depletion of Mettl3 in macrophages reshapes the tumor microenvironment (TME) by enhancing M1- and M2-like TAMs, as well as Treg infiltration into tumors. Additionally, the loss of Mettl3 activates *NF-κB* and *STAT3* signaling, resulting in the polarization of M1- or M2-like macrophages [124].

## 4. Crosstalk between Epigenetic Modifiers in Cancer Metastasis

Crosstalk between epigenetic regulators plays an important role in various biological processes such as fibrosis [145], inflammation [146], development [147], and tumor metastasis [18]. These factors, including DNA methylation, histone modifications, and ncRNAs, interact with each other in a complex and coordinated manner to modulate the epigenome [148,149]. Understanding the mechanisms and dynamics of this crosstalk is essential to elucidate the epigenetic basis of various biological processes and diseases. In the following section, we explore the fascinating area of epigenetic regulatory factor crosstalk, its significance in genomic regulation, and its impact on tumor metastasis.

Ma et al. reported that an *IFNα*-induced lncRNA negatively regulated immunosuppression by disrupting H3K27 acetylation in head and neck squamous cell carcinoma (HNSCC) (Figure 3A). Mechanistically, *lncMX1-215* directly interacted with *GCN5*, a well-known H3K27 acetylase, interrupting its binding to H3K27 for acetylation. The overexpression of *lncMX1-215* inhibited both the proliferation capacity and metastatic ability of HNSCC in vitro and in vivo [150]. In addition to lncRNAs, miRNAs are involved in regulating histone acetylation. Research has shown that *miR-15a-5p* inhibits lung cancer cell proliferation, migration, and invasion by directly targeting the 3′-UTRs of ACSS2 (Figure 3B). This leads to a reduced acetate uptake for lipid synthesis, which subsequently decreases acetyl-CoA activity and histone H4 acetylation. Furthermore, through the regulation of histone H4 acetylation, *miR-15a-5p* suppresses the transcriptional expression of the lipid metabolism genes *FASN* and *ACC*, thereby inhibiting lipid metabolism and metastasis in lung cancer [151].

Furthermore, miRNAs can be silenced via DNA methylation. A reduction in *miR-133a-3p* expression significantly enhanced the migration, invasion, proliferation, and stemness of breast cancer cells (Figure 3C). Epigenetic silencing of *miR-133a-3p* resulted in the abnormal upregulation of MAML1, which facilitated breast cancer metastasis both in vitro and in vivo. Notably, MAML1 overexpression activated the expression of DNA methyltransferase 3A (DNMT3A), thereby further increasing methylation at the miR-133a-3p promoter region. Therefore, targeting the positive feedback axis involving *miR-133a-3p/MAML1/DNMT3A* could be a potential therapeutic strategy for breast cancer [152].

Additionally, dynamic crosstalk exists between DNA methylation and histone modifications. Na et al. revealed that KMT2C deficiency promotes small-cell lung cancer (SCLC) metastasis through DNMT3A-mediated epigenetic reprogramming (Figure 3D). Specifically, KMT2C directly regulates DNMT3A expression via histone methylation and chromatin-remodeling mechanisms. DNMT3A has been identified as a tumor-suppressor gene in both leukemia and non-small-cell lung cancers. Their findings revealed that DNMT3A deficiency mediates the activation of pro-metastatic genes in PRMK-SCLC cells. Given the crucial roles played by coordinated histone methylation and DNA methylation driven by KMT2C and DNMT3A deficiencies, respectively, during SCLC metastasis progression, researchers have proposed SAM (S-adenosylmethionine), a common substrate for both histones and DNMTs, as a proof-of-concept inhibitor for SCLC metastasis treatment. Treatment with SAM significantly inhibited the growth of PRMK tumor organoids in a dose-dependent manner and reduced the formation of axon-like protrusions within these organoids [153]. Another example of methylation modification crosstalk was presented by Deng et al. (Figure 3E). They demonstrated that METTL3-mediated m6A formation in RNA causes DNA demethylation in adjacent genomic loci in both normal and esophageal squamous cell carcinoma (ESCC) cells. This process is mediated by the interaction between the m6A reader FXR1 and DNA 5-methylcytosine dioxygenase *Tet1*. By detecting m6A in RNA, FXR1 recruits Tet1 to genomic loci to demethylate the DNA, resulting in reprogrammed chromatin accessibility and gene transcription. They characterized a regulatory mechanism of chromatin accessibility and gene transcription mediated by the coupling of RNA m6A formation with DNA demethylation, highlighting the significance of crosstalk between RNA m6A and DNA modification in ESCC cell proliferation, migration, and invasion [154].

Crosstalk between DNA methylation and chromosome modification is equally important for tumor migration. ARID1A is a component of Switch/Sucrose Non-Fermentable (SWI/SNF) chromatin-remodeling complexes. Luo et al. found that ARID1A depletion drives the proliferation and metastasis of squamous cells. This is because it triggers a shift in the transcriptome from tumor suppression to carcinogenesis. Upon further investigation, it was found that the depletion of ARID1A was caused by hypermethylation of its promoter [155]. Liu et al. identified *CLDN6* as a potential breast cancer-suppressor gene [156]. Further exploration found that the methylation of CpG dinucleotides in the *CLDN6* promoter may not directly interfere with the binding ability of transcription factor, but methyl-CpG binding protein may interfere with it. To be specific, CLDN6 expression is related to DNA methylation in breast cancer tissues and MCF-7 cells. DNA methylation of *CLDN6* gene down-regulates its expression by recruiting MeCP2, deacetylating H3 and H4, and altering the chromatin structure [157].

The intricate crosstalk between epigenetic modifiers in cancer metastasis is multifaceted and involves more than three interacting epigenetic modifiers. For instance, HCC is a highly lethal neoplasm, with a 5-year survival rate of approximately 18% in patients with metastatic diseases [158]. In addition to genetic mutations, aberrant epigenetic mechanisms, including dysregulated miRNAs, significantly influence hepatocyte transformation and remodeling of the TME. Zhao identified the *miR-144/miR-451a* cluster as a crucial suppressor of HCC development. These miRNAs promote macrophage M1 polarization and exert antitumor activity by targeting hepatocyte growth factor (HGF) and macrophage migration inhibitory factor (MIF) (Figure 3F). A feedback loop exists between the *miR-144/miR-451a* cluster and EZH2, the catalytic subunit of PRC2. In HCC tissues, EZH2 binds to and induces histone H3K27me3 in the −200 to 0 bp region of the *pri-miR-144/451a* promoter. EZH2 is also a reported target of *miR-144* and is inhibited by this miRNA. The researchers further revealed that DNMT1 methylation of the CpG island upstream of *miR-144/miR451a* prevents enhancer E1 from binding to the promoter, thereby inhibiting the expression of *pri-miR-144/451a*, which is consistent with the phenomenon observed in clinical HCC samples [159].

In summary, epigenetic modifications such as DNA methylation, histone modifications, ncRNAs, and chromatin accessibility interact with each other and co-regulate gene expression and cellular functions. These interactions can generate feedback loops or crosstalk mechanisms that play important roles in tumorigenesis, metastasis, and other biological processes. Understanding these epigenetic modifications and their interactions provides new opportunities for the development of therapeutic strategies and exploration of disease mechanisms.

## 5. Clinical Prospects of Epigenetic Modifiers in Cancer Metastasis

Metastasis remains a major challenge in cancer treatment. Epigenetic regulation is a critical factor in the development and progression of cancer. Epigenetic regulation can modulate gene expression without altering DNA sequence, thereby influencing cellular phenotypes and functions. In previous sections, multiple critical findings over the last five years have highlighted the role of epigenetics as a driver of cancer metastasis.

One of the promising areas in the clinical application of epigenetic regulation is the development of epigenetic biomarkers. Epigenetic markings are closely associated with tumor type and disease stage, as well as individual genetic diversity—for example, in personalized medicine. They have the potential to provide molecular biomarkers for diagnosis and therapeutic alternatives to cancer treatment [160,161,162].

Epigenetic drugs (epi-drugs) offer a therapeutic approach that targets epigenetic modifications. Small molecules, such as DNA methyltransferase inhibitors (DNMTi) and HDACi, have been found to be beneficial in the treatment of hematological malignancies by activating silent tumor suppressors. For instance, azacitidine+ decitabine or low-dose cytarabine (Venclexta), a epigenetic compound approved by the Food and Drug Administration (FDA), have been found to treat acute myeloid leukemia by the mechanism of DNMTi. By the mechanism of HDACi, epigenetic compounds approved by the FDA, including romidepsin or vorinostat, have been used for the treatment of cutaneous T-cell lymphoma [163]. However, the results for solid tumors remain controversial [164,165]. Most clinically approved epi-drugs are currently used in non-solid tumors, such as DNMTi 5-azacytidine and 5-Aza-2-deoxycytidine for the treatment of myelodysplastic syndrome [166,167]. The first epigenetic drug applied to solid tumors, the *EZH2*-inhibitor tazemetostat, was approved in 2020 for the treatment of locally advanced or metastatic epithelioid sarcoma, highlighting the potential of epigenetic therapy [168].

Notably, combination therapies involving epigenetic modifiers and conventional anticancer agents may improve therapeutic efficacy. Numerous ongoing clinical trials are exploring other epigenetic targeted therapies for solid tumors, including combined chemotherapy, targeted therapy, or immunotherapy to enhance synergies, indicating a promising future for the development of epigenetic-targeted therapies for patients with cancer [169,170].

Moreover, epigenetic profiling can provide valuable information for predicting treatment outcomes and for monitoring disease progression. Epigenetic markers have been used to predict the chemotherapy response in patients with lung cancer, enabling personalized treatment decisions [171].

The main advantage of targeted epigenetic modification is that cancer cells are more dependent on specific epigenetic changes that occur during cancer progression than normal cells, and the reversible nature of epigenetic modification allows cancer cells to be reprogrammed back to their normal phenotype [172]. As many of these epigenetic alterations are induced by the translation of signaling pathways, new epigenetic treatments specifically designed for metastatic diseases should be developed. Although epi-drugs have the potential to inhibit metastasis progression, several key mechanistic questions and challenges remain to be addressed to fully utilize these therapies effectively and safely in clinical settings.

Despite the significant potential of epigenetic regulation in cancer metastasis, several challenges remain to be addressed. One of the main issues is the lack of specificity of first-generation epi-drugs, which are relatively non-specific, owing to the high degree of similarity among epigenetic proteins. KDM5 demethylase inhibitors designed for their active sites also inhibit other KDM5 family members (KDM5A/B/C/D), owing to the high homology of their active sites [173]. Epigenetic therapy may require long-term administration to be effective; hence, target selectivity is critical for limiting toxicity. Most in vitro experiments have focused solely on the effects of epi-drugs on tumor cells. However, in vivo administration may directly or indirectly affect the epigenetic state of normal cells, thereby increasing drug toxicity. Additionally, metastatic cancers will eventually develop resistance to exogenous drugs; therefore, epi-drugs must be specifically designed to minimize resistance and metastatic recurrence. Notably, KDM5 demethylase inhibitors have been shown to inhibit drug resistance in various cancer types [174,175,176,177].

Overall, the clinical application of epigenetic regulation in cancer metastasis holds great promise. Further investigation and clinical trials are warranted to fully comprehend the possibilities in this field, with the ultimate goal of improving the diagnosis and treatment of cancer metastasis and enhancing patient survival and quality of life. The examples provided herein demonstrate the significance and potential of epigenetic regulation in cancer metastasis and pave the way for more effective therapeutic options in the future.

## 6. Conclusions and Perspectives

The epigenetic status of a tumor can strongly influence its behavior and aggressiveness, alter various signaling pathways, and regulate metastasis-related factors such as cell adhesion molecules, the extracellular matrix, matrix metalloproteinases, metastasis-related proteins, and transcription factors. Because tumors often metastasize before diagnosis, diagnostic tools or prognostic markers are necessary to predict the risk or stage of metastasis in patients with cancer.

This review summarizes the functions of DNA methylation, histone modification, ncRNAs, chromatin accessibility, enhancer reprogramming, and m6A in cancer metastasis and highlights their molecular mechanisms. It discusses the mechanisms by which these modifiers can affect gene expression and cell behavior, ultimately contributing to tumor metastasis. Additionally, epigenetic modifiers have been identified as potential therapeutic targets for cancer treatment. The future of epigenetic modifiers in cancer metastasis holds great promise, with potential implications in improving cancer prognosis and treatment outcomes.

However, the functions of numerous epigenetic modifiers remain undetermined, and our current understanding of cancer metastasis-specific epigenetic modifiers appears to be only the tip of the iceberg, emphasizing the urgent need for basic and clinical researchers to address this issue. Cancer metastasis is a continuous process that occurs within a complex TME, which further complicates the application of epigenetic modifiers in clinical settings. It is a significant challenge for researchers to identify cancer metastasis-specific epigenetic modifiers and their targets, as well as elucidate the complex regulatory network based on epigenetic modifiers. The interactions between different epigenetic modifications and their contribution to cancer metastasis remain elusive. Epigenetic modifiers are unlikely to enter clinical practice in medical oncology as a standalone monotherapy. Instead, these drugs can be used in combination with conventional anticancer drugs, targeted therapies, and immune checkpoint inhibitors. However, the risks associated with this combination therapy should also be considered. For example, there may be interactions between drugs that affect efficacy or increase side effects. Therapeutic targets are relatively complex, and if epigenetic modifications affect multiple genes and pathways that are intertwined with the targets of traditional drugs, they can make therapeutic effects difficult to predict. In addition, due to individual differences, combination therapy may cause some patients to be unable to tolerate the therapy or have poor treatment effects. In conclusion, identifying the most promising targets for inhibiting cancer metastasis and developing novel therapeutic approaches remains challenging.

Ongoing research in this field is crucial for advancing our understanding of the potential of epigenetic modifiers in the fight against cancer metastasis. Further research is needed to fully understand these mechanisms and to develop effective treatments. Moreover, determining which should take precedence and serve as viable pharmacological targets may demand extensive collaboration among disciplines, such as artificial intelligence, molecular mechanism research, and drug development. Despite the challenges of developing epigenetic modifier-based therapeutic strategies, we believe that a new targeted therapeutic era based on epigenetic modifiers for cancer metastasis is on the horizon.

## Figures and Tables

**Figure 1 biomolecules-14-00916-f001:**
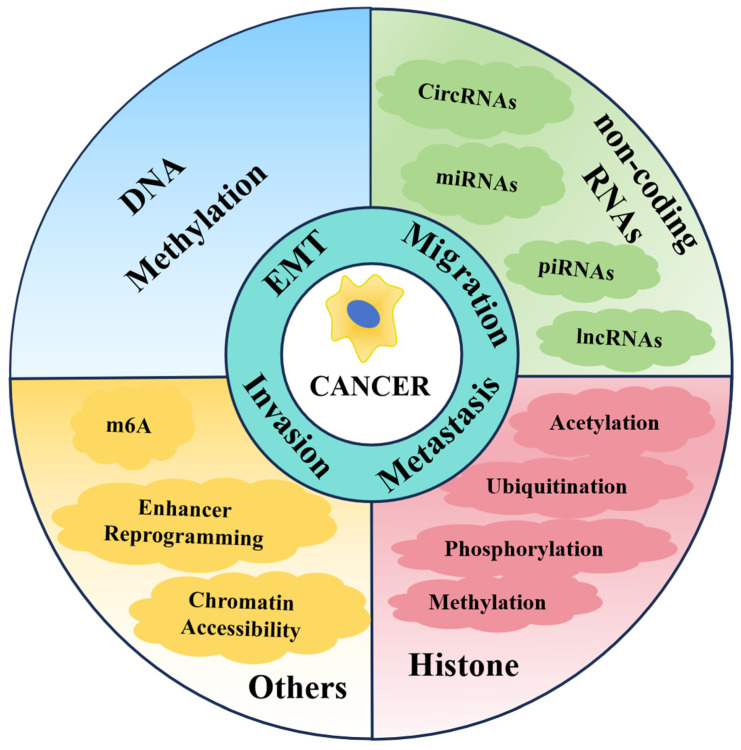
The role of epigenetic modifiers, including histone modifications, DNA methylation, non-coding RNAs, enhancer reprogramming, chromatin accessibility, and N6-methyladenosine, in metastasis-associated processes, such as epithelial-mesenchymal transition, cancer cell migration, and invasion.

**Figure 2 biomolecules-14-00916-f002:**
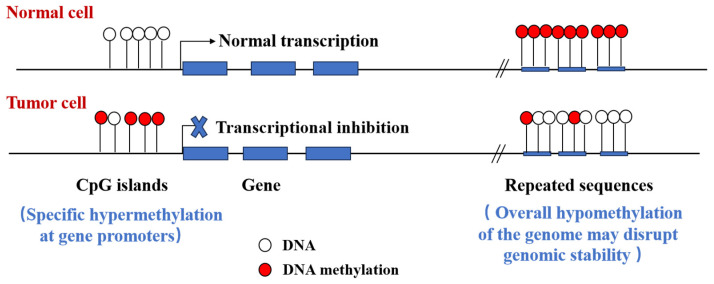
Genomic sites of DNA methylation and their effects on gene expression.

**Figure 3 biomolecules-14-00916-f003:**
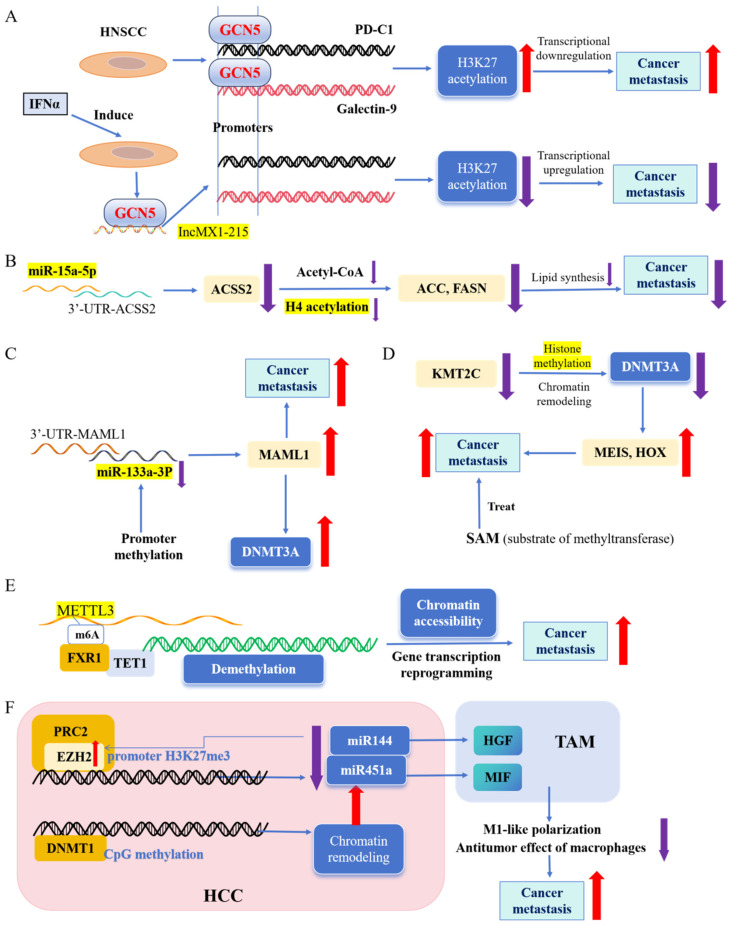
Crosstalk between epigenetic modifiers in cancer metastasis. (**A**). Novel IFNα-induced long noncoding RNA negatively regulates immunosuppression by interrupting H3K27 acetylation in head and neck squamous cell carcinoma [118]. (**B**). miR-15a-5p inhibits metastasis and lipid metabolism by suppressing histone acetylation in lung cancer [119]. (**C**). Methylation-mediated silencing of miR-133a-3p promotes breast cancer cell migration and stemness via miR-133a-3p/MAML1/DNMT3A positive feedback loop [120]. (**D**). KMT2C deficiency promotes small-cell lung cancer metastasis through DNMT3A-mediated epigenetic reprogramming [121]. (**E**). RNA m^6^A regulates transcription via DNA demethylation and chromatin accessibility [122]. (**F**). Epigenetic silencing of miR-144/451a cluster contributes to HCC progression via paracrine HGF/MIF-mediated TAM remodeling [124].

**Table 1 biomolecules-14-00916-t001:** Summary of metastasis-associated DNA methylation.

DNA Methylation	Target Genes	Pathways	Cancer Types	Location of MTS	The Role in Tumor Metastasis	Ref.
DNMT1	*PAS1*	*DNMT1-PAS1-PH20* axis	BC	Liver	Facilitator	[45]
DNMT1/DNMT3a	*B7-H3*		NEPC	Liver	Facilitator	[46]
DNMT1	*PCDH10*		GC	Lymph node	Facilitator	[47]
DNMT1	*ARID2*	*DNMT1-Snail* axis	HCC	Lung	Inhibitor	[48]
DNMTs	*NQO-1, ALDH1a3*		PDA	Liver and peritoneal	Facilitator	[49]
DNMT3B	*CTH*		HCC	Lung	Facilitator	[50]
	*CDCA3*		GC		Inhibitor	[51]
	*CPEB1*		CRC		Facilitator	[52]
DNMTs	*EMX1*	*EMX1-EGFR-ERK* axis	HCC	Lung	Facilitator	[41]
DNMT1	*Reprimo*		GC	Liver	Facilitator	[53]
DNMT3A	*CD147*		NSCLC	Lymphatic	Inhibitor	[54]
	*FAS*		OS	Lung	Facilitator	[55]
DNMT1	*miR-29b-3p*		Pancreatic cancer		Facilitator	[56]
DNMT1	*TRAF6*		CRPC		Facilitator	[57]

Malignant tumor shift (MTS); non-small-cell lung cancer (NSCLC); breast cancer (BC); neuroendocrine prostate cancer (NEPC); pancreatic ductal adenocarcinoma (PDA); gastric cancer (GC); castration-resistant prostate cancer (CRPC).

**Table 2 biomolecules-14-00916-t002:** Summary of metastasis-associated histone modifications.

Histone Modifications		Target Genes	Pathways	Cancer Types	Location of MTS	The Role in Tumor Metastasis	Ref.
Histone methylation	SMAD1	*SOX2 (H3K27)*		OC		Inhibitor	[59]
	SMAD3	*SOX2 (H3K27)*		OC		Facilitator	[59]
	SETD2	*EZH2 (H3K36)*	*SETD2-EZH2* axis	Pca		Inhibitor	[60]
	SUV39H1	*PH20 (H3K9)*	*DNMT1-PAS1-PH20* axis	BC	Liver	Facilitator	[45]
	KDM5C	*METTL14 (H3K4)*		CRC	Lymph node	Facilitator	[61]
Histone acetylation	p300	*COL6A1*		OS	Lung	Facilitator	[62]
	HDAC4	*NEDD9 (H3K9)*		BC	Lung	Inhibitor	[63]
		*CSF-1 (H3K27)*		ESCC	Lung	Facilitator	[64]
	HDAC11	*LKB1*		HCC	Lung	Facilitator	[65]
	CSRP2BP	*N-cadherin (H4)*		Cervical cancer	Lung	Facilitator	[66]
Histone ubiquitination	USP22/RNF40			OS	Lung	Facilitator	[67]
	RNF2	*E-Cadherin (H2K119)*		HCC	Lung/bone	Facilitator	[68]
Histone phosphorylation	IKKα	*NF-κb (H3)*	*HECTD3-IKKalpha* axis	BC	Lung	Facilitator	[69]
	PKM2	*DKK1*		HCC	Lung	Facilitator	[70]

Ovarian cancer (OC); prostate cancer (Pca); esophageal squamous cell carcinoma (ESCC).

**Table 3 biomolecules-14-00916-t003:** Summary of metastasis-associated ncRNAs.

ncRNAs		Target Genes	Pathways	Cancer Types	Location of MTS	The Role in Tumor Metastasis	Ref.
miRNAs	*miR-500a-5p*	*USP28*		BC	Lung	Facilitator	[81]
	*miRNA-802*	*MYLIP*		Cervical cancer		Inhibitor	[82]
	*MIR106A-5p*	*BTG3*		NPC	Lung	Facilitator	[83]
	*MiR-328-3p*	*CPT1A*	*miRNA-328-3p-CPT1A* pathway	BC	Lung	Inhibitor	[84]
lncRNAs	*ELNAT1*	*UBC9*		Bca	Lymph node	Facilitator	[85]
	*NEAT1*	*RPRD1B*	*c-Jun/c-Fos/SREBP1* axis	GC	Lymph node	Facilitator	[86]
	*LINC00926*	*PGK1*	*FOXO3A/LINC00926/PGK1* axis	BC	Lung	Inhibitor	[87]
	*PCGEM1*	*P4HA2*		GC		Facilitator	[88]
circRNAs	*circRNA*			HCC	Lung	Facilitator	[89]
	*circRNA_0000140*	*LATS2*	*miR-31/LATS2* axis of *Hippo* signaling pathway	OSCC	Lung	Inhibitor	[90]
	*circASH2*	*TPM4*		HCC	Lung/gastrointestinal tract	Inhibitor	[91]
piRNAs	*piR-19166*	*CTTN*		Pca		Inhibitor	[92]
	*piRNA-1742*	*USP8*		RCC		Facilitator	[93]
	*piR-57125*	*CCL3*	*AKT/ERK* pathway	ccRCC	Lung	Inhibitor	[94]

Nasopharyngeal carcinoma (NPC); bladder cancer (BCa); oral squamous cell carcinoma (OSCC); renal cell carcinoma (RCC); clear-cell renal cell carcinoma (ccRCC).

**Table 4 biomolecules-14-00916-t004:** Summary of metastasis-associated enhancer reprogramming, chromatin accessibility, and m6A.

Epigenetic Types		Target Genes	Pathways	Cancer Types	Location of MTS	The Role in Tumor Metastasis	Ref.
m6A		*SPRED2 mRNA*		Melanoma	Lung	Facilitator	[124]
		*Slug mRNA*		HNSCC	Lymphatic and EMT	Facilitator	[125]
		*ITGB1 mRNA*		EOC	Lymph node	Inhibitor	[126]
		*HMGA1 mRNA*		CRC		Facilitator	[127]
		*circCCDC134*		CC	Lung	Facilitator	[128]
Chromatin accessibility	*CECR2*			BC	Lung and other organs	Facilitator	[129]
	*ZBTB18*			BC	Lung	Facilitator	[130]
		*NID1*	*RUNX2/NID1* signaling	GC		Facilitator	[131]
Enhancer reprogramming	*GATA5/FOXA1*	Gain enhancer		PDA	Lung	Facilitator	[132]
	*miR-26A1*	Enhancers enrichment of *H3K27ac*		NSCLC		Inhibitor	[133]

Head and neck squamous carcinoma (HNSCC); epithelial ovarian cancer (EOC); cervical cancer (CC).

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
