# Peer review of "Epigenetic Modifiers in Cancer Metastasis"

_biomolecules, 2024, doi:10.3390/biom14080916_

Round 1

Reviewer 1 Report

Comments and Suggestions for Authors

Authors summarized up to date results about the role of epigenetic modifications in metastasis development and discussed the potential of epigenetic modifier for clinical utility, namely for diagnosis, prognosis and therapy of cancer metastatic disease.

This topic is actual, but several similar reviews were published yet.

I have several recommendation:

·        Emphasize the new message of this review in comparing to other published similar reviews as for example Janin et al., Cancer and Metastasis Reviews (2023) 42:1071–1112.

·        However, I have appreciate graphical representation of crosstalk between epigenetic modifiers in cancer metastasis (Figure 2). It clarified the relatively complicated text at paragraph 4.

·        This manuscript contains many sentences with general knowledge without any particular results. Insert the more concrete results.

·        Several longer paragraphs cite only one or two references. Insert relevant references at the right sites. There is a rule that reference need to be written after the all data, which contain this reference for clear identification of the information source.

·        Delete or reformulate sentences without any relevant data, such “cotton” in the text for example row 57 or 237.

·        Are there any other data in epigenetic therapy in hematologic malignancies or solid tumors, which where FDA approved or which are still in clinical trials?

·        Tables 1- 4 – Revise head of tables.

-        Do not use the word “Promoter” for initiation or development of cancer metastasis. Promoter is defined as regulation sequence of genes.

-        Do not the word “Biomarker” for cancer-associated changes predominately obtained from cancer cell lines. Firstly, genes, ncRNAs and named epigenetic events exist also in normal cells. The changes (up or down regulation – insert at tables!) in their expression or processes contribute to cancer development. Secondly, laboratory results could represent only potential biomarkers. Final biomarkers need to be carefully verified in larger group of patients.

-        Data in the column “Cancer types” could be divided into two columns, cancer type and location of MTS.

-        Control abbreviations under the tables.

·        Control the names of genes and proteins (italics, capitals…) in the whole text.

For this manuscript, I recommend the major revision.

Comments on the Quality of English Language

Minor revision.

Author Response

Comments 1: [Emphasize the new message of this review in comparing to other published similar reviews as for example Janin et al., Cancer and Metastasis Reviews (2023) 42:1071–1112. 

However, I have appreciate graphical representation of crosstalk between epigenetic modifiers in cancer metastasis (Figure 2). It clarified the relatively complicated text at paragraph 4.] 

Response 1: [Thank you for pointing this out. We fully agree with this valuable comment. Compared to previous literature, we have comprehensively analyzed recent studies on the molecular mechanisms and activities of epigenetic modifiers associated with cancer metastasis. Additionally, we have summarized the crosstalk between epigenetic regulators plays an important role in tumor metastasis and the clinical prospects of epigenetic modifiers in cancer metastasis. The new message of this review in comparison to other published similar reviews has been added in the abstract and the introduction. Please check Page 1 (Line 20-24) and Page 2 (Line 48-52).]

Comments 2: [This manuscript contains many sentences with general knowledge without any particular results. Insert the more concrete results.]   

Response 2: [Thank you for pointing this out. We agree with this comment. Similar comments were made by reviewer 2. We added concrete results in the relevant content. For example, how the target genes (e.g. IL6, EMX1, etc.) are involved in cancer metastasis. Please check Page 5 (Line 166-174 and 183-189).]

Itoh et al. reported that interleukin-6 (IL-6) expression in primary tumor cells is increased by the hypoxic tumor microenvironment and that increased expression may contribute to tumor metastasis. IL-6 expression in tumor cells is regulated by DNA methylation. In primary tumor cells, IL-6 expression is increased by DNA demethylation of its promoter via ten-eleven translocation 2 (Tet2). Tet2 activity is associated with increased metastatic capacity of OS cells. Tet2-dependent IL-6 induction promotes a metabolic shift andincreased lung colonization activity, enhancing metastasis in osteosarcoma (OS) cells. These findings suggest thatblocking IL-6 signaling could be serve as a potential therapy to antagonize metastasis.

EMX1 was recognized as a potential clinically available epi-marker in hepatocellular carcinoma (HCC). Wen et al. found that EMX1 gene body DNA methylation at R2–R4 positively regulates EMX1 expression in HCC. EMX1-FL, but not EMX1-X1, promotes the migration, invasion and metastasis of HCC in vivo and in vitro by activating EGFR transcription and EGFR-ERK signaling, while blocking the EGFR signals can reversed these effects and reduced HCC metastasis. Their results revealed that EMX1-EGFR is a potential target for reducing HCC metastasis.

Comments 3: [Several longer paragraphs cite only one or two references. Insert relevant references at the right sites. There is a rule that reference need to be written after the all data, which contain this reference for clear identification of the information source.]  

Response 3:  [Thank you for pointing this out. We agree with this comment. In this paper, we have inserted relevant references in Section 2 (Ref.11-13). In addition, the position of all references has been modified in accordance with the rule. We have focused on this issue in subsequent revisions. Please check in the revised version.]

Comments 4: [Delete or reformulate sentences without any relevant data, such “cotton” in the text for example row 57 or 237.]   

Response 4:  [Sorry, we didn't find this word “cotton” throughout the text. However, in this paper, we have made several revisions, such as DNA methylation (Section 3.1), non-coding RNA (Section 3.3) where we have added more detailed introduction.]

Comments 5: [Are there any other data in epigenetic therapy in hematologic malignancies or solid tumors, which where FDA approved or which are still in clinical trials?]  

Response 5:  [Epigenetic drugs offer a therapeutic approach that targets epigenetic modifications. Small molecules, such as DNA methyltransferase inhibitors (DNMTi) and HDACi, have been found to be beneficial in the treatment of hematological malignancies by activating silent tumor suppressors. For instance, azacitidine+ decitabine or low-dose cytarabine (Venclexta), a epigenetic compound approved by Food and Drug Administration (FDA), have been to treat acute myeloid leukemia by the mechanism of DNMTi. By the mechanism of HDACi, epigenetic compounds approved by FDA including romidepsin or vorinostat for the treatment of cutaneous T-cell lymphoma. However, the results for solid tumors remain controversial. Most clinically approved epi-drugs are currently used in non-solid tumors, such as DNMTi 5-azacytidine and 5-Aza-2-deoxycytidine for the treatment of myelodysplastic syndrome. The first epigenetic drug applied to solid tumors, the EZH2-inhibitor tazemetostat, was approved in 2020 for the treatment of locally advanced or metastatic epithelioid sarcoma, highlighting the potential of epigenetic therapy.

The above is a search of FDA-approved data in epigenetic therapy in hematologic malignancies or solid tumors. Please check Page 15 (Line 549-557). ]

Comments 6: [Tables 1- 4 – Revise head of tables.

  Do not use the word “Promoter” for initiation or development of cancer metastasis. Promoter is defined as regulation sequence of genes.

  Do not the word “Biomarker” for cancer-associated changes predominately obtained from cancer cell lines. Firstly, genes, ncRNAs and named epigenetic events exist also in normal cells. The changes (up or down regulation – insert at tables!) in their expression or processes contribute to cancer development. Secondly, laboratory results could represent only potential biomarkers. Final biomarkers need to be carefully verified in larger group of patients.

-   Data in the column “Cancer types” could be divided into two columns, cancer type and location of MTS.

  Control abbreviations under the tables.]

Response 6:  [Thanks for your meaningful suggestion. We fully agree with this comment. Tables 1- 4 have been revised in accordance with the above comments.]   

Comments 7: [Control the names of genes and proteins (italics, capitals…) in the whole text.]

Response 7:  [We thank for this professional comment. The names of genes and proteins have been highlighted in italics throughout the text.]

4. Response to Comments on the Quality of English Language

Point 1: [Minor editing of English language required]

Response 1: [Thank you for your suggestion of minor editing of English language to help us improve the quality of our manuscripts. We have checked and corrected linguistic errors such as spelling and syntax throughout the manuscript. However, If you suggest that our revised manuscript still needs English language editing, we will take it further.] 

5. Additional clarifications

[Here, mention any other clarifications you would like to provide to the journal editor/reviewer.]

Reviewer 2 Report

Comments and Suggestions for Authors

The manuscript by Hu et al is a review of epigenetic modifiers in cancer metastasis. It covers DNA methylation, chromatin modifications, non-coding RNAs, enhancer reprogramming. It is an informative, well written, and up to date review. However, it is overly broad and the significance of the examples presented is not always clear. The discussion could benefit from more extensive discussion of the key epigenetic players and more description of the target genes. Here are specific suggestions for improvement:

1.      DNA methylation section: It would be helpful if this section could start with a description of DNA methyltransferases, Tet enzymes, and other key regulators of DNA methylation by discussing what they are, enzymatic activities. In addition, I would suggest adding information about the genomic distribution of DNA methylation (e.g. define CpG islands, shores, and discuss the importance of DNA methylation at repetitive genomic regions such as retrotransposons and their significance in cancer metastasis. It would be helpful to have a figure showing genomic sites of DNA methylation and their effects on gene expression.

2.      The significance of the genes regulated by DNA methylation is not clear. It would help to explain how the target genes (e.g. IL6, EMX1 etc.) are involved in cancer metastasis.

3.      It would be good to include a discussion of methylation at repetitive sequences.

4.      For the histone modifications, there should be an introduction of the different histone modifications, the enzymes involved,  and their effects on transcription. I suggest adding a figure as well.

5.      Similar to DNA methylation, the function of the target genes such as Nedd9 should be explained in more detail.

6.      Additional detail should be added to the introductory paragraphs on micro, lnc, and circular RNAs.

7.      Explain in more detail, general aspects of exosomal non-coding RNAs.

8.      It would be helpful to include a table with the different epigenetic therapeutics, their targets, and the cancers they are used in.

9.      In section 4 on crosstalk, it would be helpful to add how DNA methylation and chromatin modifications are specifically coordinated, e.g. by methyl DNA binding proteins such as MECP2.

10.   Section 5 on clinical prospects: expand on the statement in lines 499-500 on the risks associated with combination therapy. What are potential risks?

11.   Please define all abbreviations. E.g. It is not clear what “OS” refers to.

Author Response

Comments 1: [DNA methylation section: It would be helpful if this section could start with a description of DNA methyltransferases, Tet enzymes, and other key regulators of DNA methylation by discussing what they are, enzymatic activities. In addition, I would suggest adding information about the genomic distribution of DNA methylation (e.g. define CpG islands, shores, and discuss the importance of DNA methylation at repetitive genomic regions such as retrotransposons and their significance in cancer metastasis. It would be helpful to have a figure showing genomic sites of DNA methylation and their effects on gene expression.]

Response 1[We thank for this comment. According to the valuable suggestion, we have added two paragraph and a figure to illustrate this question in Section 3.1 (Page3-5)

 The following is a brief description of the key elements involved in the DNA methylation process - DNA methyltransferase, methylated CpG adhesion protein, Tet enzymes (Ten-eleven translocation enzymes). DNA methyltransferases are a class of enzymes capable of adding methyl (-CH) groups to specific bases of DNA molecules [26]. According to the different function and role period, it is mainly divided into two categories. DNMT1: Maintenance methyltransferase, mainly in the process of DNA replication, the newly synthesized daughter DNA corresponding to the parent chain methylation site of cytosine (C) methylation modification, in order to maintain the stable inheritance of DNA methylation pattern [27]. DNMT3A and DNMT3B: de novo methyltransferases, which can catalyze new methylation at the unmethylated CpG site and establish the initial DNA methylation pattern [28]. DNA methyltransferase activity is regulated by a variety of factors, including interactions with other proteins, intracellular metabolites, transcription factors, and chromatin structure. Methylated CpG adhesion proteins are a class of proteins that can specifically recognize and bind to methylated CpG sites. By binding to methylated DNA, they recruit other protein complexes, such as histone modifying enzymes, chromatin remodeling factors, etc., to form inhibitory chromatin structures, thereby inhibiting gene transcriptional expression [29]. Tet enzymes are a class of dioxygenases that catalyze the oxidation of 5-methylcytosine (5mC) to 5-hydroxymethylcytosine (5hmC), 5-aldehyde cytosine (5fC) and 5-carboxycytosine (5caC). The activity of the Tet enzyme is essential for the process of DNA demethylation. The 5hmC, 5fC and 5caC catalyzed by them can be removed by the Base Excision Repair (BER) pathway or by deamination, thus achieving DNA demethylation [30].

DNA methylation displays a distinct distribution pattern across the genome.   CpG islands are genomic regions abundant in cytosine (C) -guanine (G) dinucleotides, primarily located in gene promoter regions , most of which remain unmethylated under normal physiological conditions, thereby facilitating the expression of associated genes [31]. The CpG shore refers to the region located within approximately 2kb upstream and downstream of the CpG Island. In certain instances, alterations in the methylation status of the CpG shore also impact the transcriptional activity of neighboring genes [32]. TEs and other forms of repetitive DNA, constituting more than half of the human genome, are the largest contributor to human genetic variation and affect human health  owing to their roles in deleterious copy number variants (CNVs), structural variants (SVs), insertions, deletions, and alterations to gene transcription and splicing [33, 34]. DNA methylation plays a crucial role in repetitive genomic regions such as retrotransposons by methylating these regions and repressing their transcription and transposition activities, thus maintaining genomic stability. At Pancreatic ductal adenocarcinoma (PDAC), Espinet found that Methylationlow tumors are characterized by higher expression of endogenous delivered transcripts and findings dsRNA sensors which leads to a cell intrinsic activation of an interferon signature (IFNsign). This results in a pro-tumorigenic microenvironment and poor patient outcome [35]

Figure 2. Genomic sites of DNA methylation and their effects on gene expression.

Aberrant changes in DNA methylation distribution patterns occur in cancer, including hypermethylation in the tumor suppressor gene promoter region's CpG island and hypomethylation in repeat sequence regions. These modifications significantly affect gene expression and genome stability, exerting a critical influence on cancer initiation, progression, and metastasis [36, 37](Figure 2). ]

Comments 2: [The significance of the genes regulated by DNA methylation is not clear. It would help to explain how the target genes (e.g. IL6, EMX1 etc.) are involved in cancer metastasis.]

Response 2: [We fully agree with this comment. It has been added in Page 5 (Line 166-174 and 183-189).

Itoh et al. reported that interleukin-6 (IL-6) expression in primary tumor cells is increased by the hypoxic tumor microenvironment and that increased expression may contribute to tumor metastasis. IL-6 expression in tumor cells is regulated by DNA methylation. In primary tumor cells, IL-6 expression is increased by DNA demethylation of its promoter via ten-eleven translocation 2 (Tet2). Tet2 activity is associated with increased metastatic capacity of OS cells. Tet2-dependent IL-6 induction promotes a metabolic shift andincreased lung colonization activity, enhancing metastasis in osteosarcoma (OS) cells. These findings suggest thatblocking IL-6 signaling could be serve as a potential therapy to antagonize metastasis [38].

EMX1 was recognized as a potential clinically available epi-marker in hepatocellular carcinoma (HCC) [40]. Wen et al. found that EMX1 gene body DNA methylation at R2–R4 positively regulates EMX1 expression in HCC. EMX1-FL, but not EMX1-X1, promotes the migration, invasion and metastasis of HCC in vivo and in vitro by activating EGFR transcription and EGFR-ERK signaling, while blocking the EGFR signals can reversed these effects and reduced HCC metastasis. Their results revealed that EMX1-EGFR is a potential target for reducing HCC metastasis [41].]

Comments 3: [It would be good to include a discussion of methylation at repetitive sequences.]

Response 3: [We fully agree with this comment. It has been added in Page 5 (Line 195-202).

Repetitive elements (REs) compose about 50% of the human genome and are normally transcriptionally silenced [43], although the mechanism has remained elusive. Shen et al. identified FBXO44 as an essential repressor of REs in cancer cells. FBXO44 bound H3K9me3-modified nucleosomes at the replication fork and recruited SUV39H1, CRL4, and Mi-2/NuRD to transcriptionally silence REs post-DNA replication. FBXO44/SUV39H1 are crucial repressors of RE transcription, and their inhibition selectively induces DNA replication stress and viral mimicry in cancer cells [44].]

Comments 4: [For the histone modifications, there should be an introduction of the different histone modifications, the enzymes involved,  and their effects on transcription. I suggest adding a figure as well.]

Response 4: [Thanks for the valuable suggestion. Histone modifications, including methylation, acetylation, ubiquitination, and phosphorylation, are critical for regulating oncogenes and tumor suppressor genes at the transcriptional level during tumor progression. The different histone modifications, the enzymes involved, and their effects on transcription were introduced in Section 3.2 by detailed examples and Table2. In addition, considering the length of the article and other factors, we did not add another graph to represent.]

Comments 5: [Similar to DNA methylation, the function of the target genes such as Nedd9 should be explained in more detail.]

Response 5: [Thanks for the Meaningful suggestion. The relevant content such as Nedd9, CDH5, PTBP1 has been added on Page 7 (Line 235-243 and 217-219) and Page 10 (Line364-367).

NEDD9, also named as casl and hef1, encodes a multi-domain scaffolding protein involved in cell signaling, and is known to regulate multiple cellular processes, such as cells proliferation, DNA damage response, migration, and so on [74, 75]. Some studies have confirmed that NEDD9 stimulates breast cancer cells invasion by influencing EMT, and activating MMP [76].

Chromodomain helicase DNA-binding protein 5 (CHD5) is recognized as a tumor suppressor gene in cancer that controls proliferation, apoptosis, and senescence via the p19(Arf)/p53 pathway [71].

As reported by Takahashi et al., PTBP1 plays a crucial role in the invasion of CRC cells, and these invasive properties arise partially through splicing CD44 [120]. Moreover, PTBP1 has been reported to mediate the pyruvate kinase (PKM) isotype switch to facilitate the Warburg effect in CRC [121]]

Comments 6: [Additional detail should be added to the introductory paragraphs on micro, lnc, and circular RNAs.]

Response 6: [Thank you for pointing this out. We fully agree with this valuable comment. It has been added in Section 3.3 (Page8-10).

MicroRNAs (miRNAs) are ncRNAs with lengths ranging from approximately 19–22 nucleotides that regulate gene-expression post transcriptionally. They regulate gene expression mainly by targeting specific recognition sites in the 3′ untranslated region (UTR), leading to mRNA degradation [95]. Dysregulation of miRNAs is associated with the development and progression of cancer. For example, in gastric cancer, miR-20a-5p, and miR-221 elevate in both plasma and serum samples. Changes of miR-221-3p, miR-378c-3p, and miR-744-5p in serum can aid the prediction of gastric cancer 5 years before any clinical symptoms appear [96]. Furthermore, the expression of miRNA-21-5p in plasma can identify colorectal cancer patients with 90% sensitivity and specificity [97]. Notably, they play pivotal roles in the regulation of multiple cellular functions including cell proliferation and migration, and EMT phenotype. miRNAs have emerged as potential therapeutic targets for the progression of various tumor metastases.

Long non-coding RNAs (lncRNAs) are a group of RNA molecules with lengths greater than 200 nucleotides. A majority of lncRNAs are transcribed by RNA polymerase II and matured by 5' capping, 3' cleavage and polyadenylation and splicing [101]. Unlike mRNAs, few lncRNAs are universally expressed in tissues. Instead, most are expressed specifically in specific conditions and tissues, meaning functional relevance [102-104]. They play a key role in the regulation of cellular processes such as genome integrity, chromatin organization, gene expression, translation regulation, and signal transduction [105]. Compared to miRNAs, lncRNAs exhibit a more diverse mechanism for regulating gene expression [106]. In recent years, lncRNAs have been implicated in various biological processes, including tumor proliferation, migration, invasion, and angiogenesis [107]. In addition, there is evidence that exosomes are closely related to tumor metastasis, and exosomes can arrive at the site of metastasis before tumor cells, change the microenvironment, activate the biological function of cells, and thus promote tumor metastasis [81]. Exosomes contain a large number of molecules including lncRNAs. This protects lncRNAs from degradation [108]. In this review, we have outlined the molecular mechanisms by which lncRNAs contribute to cancer metastasis. Metastasis-associated lung adenocarcinoma transcript1 (MALAT1) is a conserved lncRNA in mammals that is highly abundant in many cancers [109]. Transforming growth factor beta (TGF-β)-induced upregulation of MALAT1 enhances cancer metastasis, which  is mediated by an interaction with suz12 at the transcriptional level to alter downstream events. Instead, MALAT1 silencing reduces cell proliferation and invasion and increased apoptosis by reducing MALAT1-EZH2-mediated target silencing at the epigenetic level [110].

Circular RNAs (circRNAs) are characterized by covalently closed loops generated through a process called back-splicing, resulting in circular structures that lack 5’ caps and 3’ poly (A) tails [112]. Recent studies on circRNAs in cancer have revealed that they are significantly involved in cancer initiation, proliferation, drug resistance, and most importantly, cancer metastasis [113-116]. One of the primary mechanisms by which circRNAs influence tumor metastasis is through their regulation of gene expression at the transcriptional or post-transcriptional level. For instance, circRNAs bind to miRNAs through a regulatory mechanism in which endogenous RNAs compete to indirectly regulate the expression of mRNA corresponding to downstream target genes of miRNAs, contributing to the progression of breast cancer [117]. In addition, circRNAs can also interact with proteins to affect their functions and signaling pathways related to tumor metastasis. For example, certain circRNAs can bind to proteins involved in EMT. Through these interactions, circRNAs can regulate EMT and subsequently impact the metastatic behavior of tumor cells [118]. Furthermore, circRNAs can be secreted into the extracellular space and enter the circulation, functioning as potential biomarkers for tumor metastasis. Detection of specific circRNAs in the blood or other body fluids of cancer patients may provide valuable information for the early diagnosis and prognosis of metastatic tumors, as well as for monitoring the response to therapy [119].]

Comments 7: [Explain in more detail, general aspects of exosomal non-coding RNAs.]

Response 7: [Thanks for the valuable suggestion. The relevant content has been added on Page 5 (Line 315-319).

 In addition, there is evidence that exosomes are closely related to tumor metastasis, and exosomes can arrive at the site of metastasis before tumor cells, change the microenvironment, activate the biological function of cells, and thus promote tumor metastasis [81]. Exosomes contain a large number of molecules including lncRNAs. This protects lncRNAs from degradation [108].]

Comments 8: [It would be helpful to include a table with the different epigenetic therapeutics, their targets, and the cancers they are used in.]

Response 8: [Thanks for the professional suggestion. We've considered this idea before, however, the different epigenetic therapeutics, their targets, and the cancers have been shown in Table1-4. The third paragraph of Section 5, epigenetic drugs in clinical practice have been supplemented. In addition, considering the length of the article and other factors, we did not add a new table to represent.

Epigenetic drugs offer a therapeutic approach that targets epigenetic modifications. Small molecules, such as DNA methyltransferase inhibitors (DNMTi) and HDACi, have been found to be beneficial in the treatment of hematological malignancies by activating silent tumor suppressors. For instance, azacitidine+ decitabine or low-dose cytarabine (Venclexta), a epigenetic compound approved by Food and Drug Administration (FDA), have been to treat acute myeloid leukemia by the mechanism of DNMTi. By the mechanism of HDACi, epigenetic compounds approved by FDA including romidepsin or vorinostat for the treatment of cutaneous T-cell lymphoma. However, the results for solid tumors remain controversial. Most clinically approved epi-drugs are currently used in non-solid tumors, such as DNMTi 5-azacytidine and 5-Aza-2-deoxycytidine for the treatment of myelodysplastic syndrome. The first epigenetic drug applied to solid tumors, the EZH2-inhibitor tazemetostat, was approved in 2020 for the treatment of locally advanced or metastatic epithelioid sarcoma, highlighting the potential of epigenetic therapy.

The above is a search of FDA-approved data in epigenetic therapy in hematologic malignancies or solid tumors. Please check Page 15 (Line 549-557). ]

Comments 9: [In section 4 on crosstalk, it would be helpful to add how DNA methylation and chromatin modifications are specifically coordinated, e.g. by methyl DNA binding proteins such as MECP2.]

Response 9: [Thanks for the valuable suggestion.The crosstalk example of DNA methylation and chromatin modifications has been added on Page 13 (Line 490-502).

Crosstalk between DNA methylation and chromosome modification is equally important for tumor migration. ARID1A is a component of Switch/Sucrose Non-Fermentable (SWI/SNF) chromatin-remodeling complexes. Luo et al. found that ARID1A depletion drives proliferation and metastasis of squamous cells. This is because it triggers a shift in the transcriptome from tumor suppression to carcinogenesis. Upon further investigation, it was found that the depletion of ARID1A was caused by hypermethylation of its promoter [155]. Liu et al. identified Claudin-6 as a potential breast cancer suppressor gene [156]. Further exploration found that the methylation of CpG dinucleotides in the CLDN6 promoter may not directly interfere with the binding ability of transcription factor, but methyl-CpG binding protein may interfere with it. To be specific, CLDN6 expression is related to DNA methylation in breast cancer tissues and MCF-7 cells. DNA methylation of CLDN6 gene down-regulates its expression by recruiting MeCP2, deacetylating H3 and H4, and altering chromatin structure [157]. ]

Comments 10: [Section 5 on clinical prospects: expand on the statement in lines 499-500 on the risks associated with combination therapy. What are potential risks?]

Response 10: [Thanks for the professional suggestion. The relevant content has been added on page 16 (Line 631-636).

For example, there may be interactions between drugs that affect efficacy or increase side effects. Therapeutic targets are relatively complex, and if epigenetic modifications affect multiple genes and pathways that are intertwined with the targets of traditional drugs, they can make therapeutic effects difficult to predict. In addition, due to individual differences, combination therapy may cause some patients to be unable to tolerate or have poor treatment effect.]

Comments 11: [Please define all abbreviations. E.g. It is not clear what “OS” refers to.]

Response 11: [Thanks for the suggestion. “OS” refers to osteosarcoma” on Page 5 (Line 172). All abbreviations have been defined throughout the manuscript.]

4. Response to Comments on the Quality of English Language

Point 1: [English language fine. No issues detected]

Response 1: [Thank you for your recognition]

5. Additional clarifications

[Here, mention any other clarifications you would like to provide to the journal editor/reviewer.]

Round 2

Reviewer 2 Report

Comments and Suggestions for Authors

All concerns have been addressed in the revised manuscript.